# Transcutaneous Pulsed RF Energy Transfer Mitigates Tissue Heating in High Power Demand Implanted Device Applications: In Vivo and In Silico Models Results

**DOI:** 10.3390/s22207775

**Published:** 2022-10-13

**Authors:** Mohammad L. Karim, Antonio M. Bosnjak, James McLaughlin, Paul Crawford, David McEneaney, Omar J. Escalona

**Affiliations:** 1Nanotechnology & BioEngineering Research Centre, School of Engineering, Ulster University, Newtownabbey BT37 0QB, UK; 2Paul Crawford Veterinary Services, Larne BT40 3RW, UK; 3Cardiovascular Research Unit, Craigavon Area Hospital, Portadown, Craigavon BT63 5QQ, UK

**Keywords:** transcutaneous wireless power supply, left ventricular assist devices (LVAD), medical implants, skin pulsed heating, radiofrequency power loss emulator (PLE), subcutaneous thermal profile, porcine model, cadaver model, TWESMI concept, Galvani concept

## Abstract

This article presents the development of a power loss emulation (PLE) system device to study and find ways of mitigating skin tissue heating effects in transcutaneous energy transmission systems (TETS) for existing and next generation left ventricular assist devices (LVADs). Skin thermal profile measurements were made using the PLE system prototype and also separately with a TETS in a porcine model. Subsequent data analysis and separate computer modelling studies permit understanding of the contribution of tissue blood perfusion towards cooling of the subcutaneous tissue around the electromagnetic coupling area. A 2-channel PLE system prototype and a 2-channel TETS prototype were implemented for this study. The heating effects resulting from power transmission inefficiency were investigated under varying conditions of power delivery levels for an implanted device. In the part of the study using the PLE setup, the implanted heating element was placed subcutaneously 6–8 mm below the body surface of in vivo porcine model skin. Two operating modes of transmission coupling power losses were emulated: (a) conventional continuous transmission, and (b) using our proposed pulsed transmission waveform protocols. Experimental skin tissue thermal profiles were studied for various levels of LVAD power. The heating coefficient was estimated from the porcine model measurements (an in vivo living model and a euthanised cadaver model without blood circulation at the end of the experiment). An in silico model to support data interpretation provided reliable experimental and numerical methods for effective wireless transdermal LVAD energization advanced solutions. In the separate second part of the study conducted with a separate set of pigs, a two-channel inductively coupled RF driving system implemented wireless power transfer (WPT) to a resistive LVAD model (50 Ω) to explore continuous versus pulsed RF transmission modes. The RF-transmission pulse duration ranged from 30 ms to 480 ms, and the idle time (no-transmission) from 5 s to 120 s. The results revealed that blood perfusion plays an important cooling role in reducing thermal tissue damage from TETS applications. In addition, the results analysis of the in vivo, cadaver (R1Sp2) model, and in silico studies confirmed that the tissue heating effect was significantly lower in the living model versus the cadaver model due to the presence of blood perfusion cooling effects.

## 1. Introduction

Heart failure (HF) is a global health issue and remains a growing public health problem. It remains one of the leading global causes of death [1,2,3]. Advances in cardiovascular therapeutics and implantable defibrillators (ICDs) have improved survival in heart failure. Despite these advances, patients with end-stage heart failure have limited options, and in selected cases may require cardiac transplantation. A limited availability of donor hearts restricts transplant numbers to around 4000 per year worldwide. Left ventricular assist devices (LVADs) are implanted mechanical pumps used in patients with advanced heart failure to support the circulation until a donor heart becomes available (bridge to transplant). In recent years, due to lack of donor availability, LVADs have been used as definitive therapy (destination therapy) [4,5,6,7].

LVADs are small mechanical pumps connected to the main pumping chamber of the heart (left ventricle) directing a portion of the blood flow from this chamber directly to the aorta. LVADs thus assist the heart’s pumping function and have significantly improved the survival in end-stage heart failure [5]. However, LVAD power demand is high, usually 5–30 W. This requires an external power source connected to the LVAD via a percutaneous driveline. The driveline frequently becomes infected, resulting in recurrent hospital admissions and complications including life-threatening sepsis, LVAD failure and premature death [8,9,10,11,12,13,14].

Wireless power transmission (WPT) across the skin (transcutaneous) could eliminate the use of the driveline. A suitable WPT solution technology would address driveline infections, reduce associated hospital admissions, improve the patient’s quality of life and extend the LVAD lifespan. This has been attempted previously, but the high power transmission requirement results in significant heating effect and thermal skin injury, limiting the application of this this technology [15,16,17,18,19,20]. Nevertheless, the main blocking issue to a clinically adopted solution with current WPT technology are the significant heating effects in the skin tissue around the radiofrequency (RF) coupling elements of the WPT system. These are due to power losses and/or exposure to electromagnetic (EM) fields and lead to local skin tissue damage. The heating effect in the subcutaneous receiver element (coil) of WPT for LVAD demands exceeding 5 W can reach prohibitive levels exceeding 2 °C above the baseline body temperature, leading to irreversible thermal tissue injury. The development of versatile and robust WPT technologies has attracted attention in the last two decades for implanted medical devices (IMDs) [21,22,23]. In addition to thermal injury existing WPT systems generate electromagnetic (EM) fields interacting with the living tissue, characterised by the specific absorption rate (SAR) metric, and which may cause cellular damage [24,25,26,27]. In our previous studies, a Transcutaneous Wireless Energy System for Medical Implants (TWESMI) concept, with a novel high-energy pulsed RF transmission waveform for mitigating tissue thermal effects, has been presented as a multichannel WPT proprietary system [28] (see Patents section in this article), and proposed as a solution for driving high-power rated LVADs [28,29]. A multichannel system approach offers several advantages: (a) to reduce the overall coupling energy density (a 4-channel system would present less skin heat for a fixed load power level), (b) to offer higher operational reliability by redundancy (in case a transfer channel fails), and (c) to enable capacitive coupling (using plates; not coils) [17], which is only possible with an even number of channels (minimum of two channels). The proprietary TWESMI concept includes the use of thin, compact, flexible, biocompatible flat spiral coil elements [28,29].

In this article, we present novel instrumentation developed for acquiring experimental in vivo evidence of skin thermal profiles data in a porcine model, for a comparative assessment between different WPT operating modes and states. A separate in silico modelling tool was developed to understand the factors involved in mitigating skin tissue heating effects. In particular we wished to investigate and model the effect of skin tissue blood flow on cooling the implanted coil and thus mitigating thermal injury. We thus developed a Power Loss Emulating (PLE) system for WPT associated inefficiencies heating effects when driving LVADs at different power rate levels. The study presented here complements previous work [28] and provides reliable experimental (in vivo) and numerical (in silico) methods for the development of advanced WPT systems.

In this article we (1) Describe the PLE system design, (2) Compare the thermal profiles during pulsed and continuous energy transmission protocols in vivo and (3) report thermal profiles from in silico models.

## 2. Materials and Methods

This section describes the prototype Power Loss Emulation (PLE) system, used for both the novel pulsed (TWESMI) transmission waveforms [28] and the conventional continuous transmission modes [21], the experimental setup, and the protocol design for studying the thermal effects on in vivo measurements (porcine model skin tissue), both on the alive and cadaver (no blood circulation) model states, within the same pig case. For the supporting in silico model, the geometry, multiphysics and solver setup to simulate the temperature inside the subcutaneous tissue due to RF power transfer between the primary and implanted coil are described. Note that for this study, a 2-channel PLE and a 2-channel WPT prototype system were implemented and used to record the skin temperature profile data for the in vivo and cadaver states, for both pulsed and continuous transmission modes of operation. The 2-channel systems approach for the in vivo studies would offer the additional advantages of (a) increasing (doubling) the independent experimental data gathering for skin tissue thermal profile characterisation; thus, a set of 6 pig cases (during one week) could provide the same experimental data as of 12 pig cases (requiring more resources, and during two weeks) with 1-channel WPT systems and in a shorter research time, and (b) providing a real experimental in vivo setting for validating the TWESMI multichannel concept, with each of the two channels at different and distant skin areas in the body, and delivering electric power to a single device load (LVAD electrical model).

### 2.1. Power Loss Emulation

To investigate the heating effect resulting from inefficiency in electromagnetic power transmission between transcutaneous coupling elements for particular load power delivery levels in the in vivo model blood perfusion thermal profile study experiments, a power loss emulation (PLE) system prototype was developed to carry out the porcine model (with pigs) trials. Measurements of emulated power loss heating were made in vivo and in the cadaver stages of the same animal. The measurements were obtained from ohmic heating probe elements with the same shape and area as the coils used in WPT systems, for both pulsed heating and continuous heating modes for in vivo skin tissue thermal profile studies. A block diagram of the PLE system is illustrated in Figure 1. Each channel has a pair of probes: an external side probe (coil-shaped ohmic heater element and temperature sensors array) and the implanted side probe (coil-shaped ohmic heater element and temperature sensors array). The in-house built PLE system was designed to provide two channels (Ch.1 and Ch.2) for this porcine model experimental study.

The PLE system was used to investigate the heating effects resulting from various power loss levels independently of the wireless power supply coupling method being used and their associated inefficiency, thus enabling analysis and modelling of the skin tissue thermal profile data under a wide range of power loss heating intensities with the following settings: ON-pulse power level (55–700 W), ON-pulse durations (30–480 ms), and blood perfused OFF-time cooling durations (5–120 s). Thus, the implemented PLE system enabled the study of the heating effects for both: (a) our novel TWESMI pulsed transmission protocols and (b) a conventional continuous transmission mode under equal power delivery level. This enabled a comparative assessment of the heating coefficient metrics yielded by the temperature data from the subcutaneous resistor heating element (emulated RF coupling element) both in the living model and in the cadaver (control) model of the same animals.

**Figure 1 sensors-22-07775-f001:**
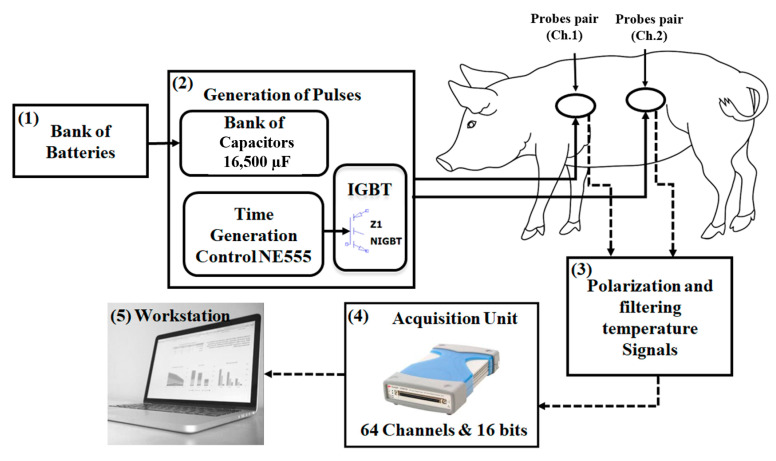
Schematic block diagram of the PLE system.

### 2.2. Wireless Power Transmission (WPT) System

The WPT system circuit layout implemented for this study is presented in Figure 2. It has and RF (set to operate @ 200 kHz) flexi-thin transmitter (Tx) side probe (coil L1 and temperature sensors array) and a flexi-thin receiver (Rx) side probe (coil L2 and temperature sensors array) with provisions for two channels (Ch.1 and Ch.2) for the porcine model experimental study. On the transmitter side, the first component is the energy source that provides the necessary voltage for each short pulse of high energy transmission and consists of 14 batteries, each having an estimated average indicative value of 12.5 volts during the time of the experiment. The output voltages from this energy source are 50 V, 75 V, 100 V, 125 V, 150 V, and 175 V. The second component includes a microcontroller for timing the pulses of energy transmission (ON state) which generates and controls the high-energy pulse width and transmission waveform duty cycle.

Table 1 presents the pulse width, idle time, and voltage levels used in the animal trials for the pulsed transmission protocol design.

**Table 1 sensors-22-07775-t001:** List of possible RF pulsed transmission mode waveform parameters protocol settings: pulse amplitude, pulse width and idle time (no-transmission for allowed idle cooling time).

Pulse Width (ms) (ON)	Idle Time (s) (OFF)	Voltage (V)
30	5	50
60	10	75
90	20	100
160	40	125
320	80	150
480	120	175

**Figure 2 sensors-22-07775-f002:**
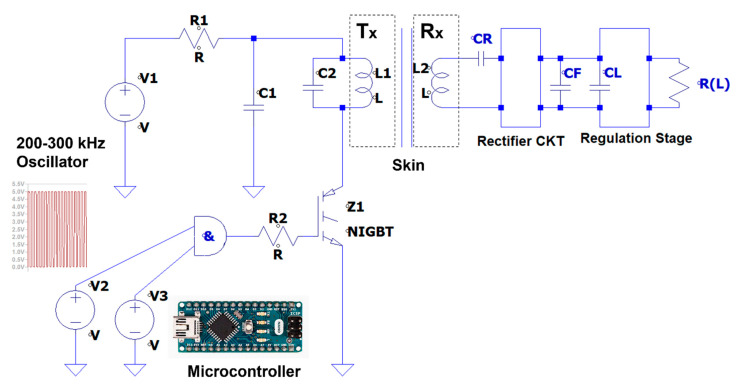
WPT system architecture with external transmitter (Tx) and implanted receiver (Rx) coil and temperature-sense-probe units.

The WPT transmission circuit utilises an IGBT as the switching power transistor. Thus, the IGBT drives the current applied directly to the primary coil (L1). The energy stored in the capacitor (C1) of the Tx side delivers the RF pulsed high currents into L1 and transfers the electromagnetic energy into the secondary coil (L2) at the 200 kHz resonate operating frequency (double-tuned coupling coils via RF capacitors C2 and CR). More specifically, the WPT system consists of the following four components:The primary side resonant tank circuit (C2 and L1): the resonant tank includes the primary coil (L1) and the tuned capacitor (C2). Therefore, it can transfer the maximum power at the selected resonance frequency. The value of the Tx tank capacitor is 10 nF, and the inductance of the Tx coil L1 is 39.74 µH.Energy Capacitor (C1): The energy capacitor delivers the stored energy into the Tx side resonant LC tank. While the IGBT is ON, all the stored energy of this capacitor is passed to the L1 coil. When the IGBT is OFF, there is no current through the IGBT transistor; however, the coil stores energy and the continuous current flowing by the C2 capacitor at the resonant frequency. The resonance frequency was set to 200 kHz.The secondary (implanted) side resonant tank circuit (CR and L2): the Rx resonant tank includes the secondary coil (L2) and the tuned capacitor (CR), also operating at 200 kHz to make the coupling system a double-tuned system. The secondary tank circuit is followed by ripple filtering stage and voltage regulation, before delivering power to the LVAD resistive model load (RL) of 50 Ω.

### 2.3. Temperature Sensors Data Acquisition Unit

The third block in Figure 1 (3) is the same for both the PLE and the WPT systems. The unit provides the polarisation and filtering functions. The unit has 64 passive RC circuits to power and filter the thermistors sensors signals acquired from the data acquisition system (block 4 in Figure 1). The multifunctional digitization unit U2356A (National Instruments, Austin, Texas) was used to implement the Analog-to-Digital (A/D) conversion for data acquisition in each one of the system’s channels to where the thermistors array probes were connected (two probes per channel, see Figure 1). The maximum sampling rate of the DAQ (U2356A) unit is 500 kSample/s, with a 16-bit resolution. The unit’s digital output is connected to a workstation to visualise the acquired thermistor sensor signals.

### 2.4. Instantaneous Pulsed Power and Energy Calculation

The peak power is defined by:(1)$$P_{o}=max\left[p\left(t\right)\right]$$

The peak power is not always readily measurable, and an instrument commonly measures the average power ($P_{avg}$). The energy per pulse can be defined as:(2)$$E_{pulse}=\int_{0}^{T}p\left(t\right)dt$$

Then the average power is:(3)$$P_{avg}=\frac{1}{T}\int_{0}^{T}p\left(t\right)dt=\frac{E_{pulse}}{T}$$
where *T* is the total period of generated pulsed RF waveform signals, that is, the time of the active transmission pulse (ON) plus the idle (OFF) time with no energy transmission. The $P_{avg}\;$ is referred to as the LVAD power requirement for a particular cardiovascular need.

For the PLE system operation, the emulated instantaneous power loss rate, can be calculated for the voltage (*V*_0_) settings of 50 V, 75 V, 100 V, 125 V, 150 V and 175 V when applied across the heating resistor ($R_{coil}$) in each of the probe pairs with a value of 45 Ω. Equation (4) is used to estimate the instantaneous emulated power loss (*P*_0_) with the PLE system. Table 2 presents the calculated values of PLE emulated instantaneous power loss levels using Equation (4).
(4)$$P_{0}=\frac{V_{0}^{2}}{R_{coil}}$$

### 2.5. Continuous Transmission Mode Equivalent DC Voltage Calculation

The equivalent low-level DC voltage at the secondary side load (RL) required for the continuous transmission mode was estimated from Equation (5). The equivalent supply voltage at the primary side was derived from the same power supply applied during the RF pulsed energy transmission (ON time) by dividing the total energy pack delivery per RF transmission pulse (ON time) by the entire period. The total period is the sum of the time the pulse is active and the idle time. Moreover, channel 1 and channel 2 of the implanted coil (Rx) are connected to a dummy load after the rectification, filtering and regulation stages of the LVAD model resistance value of 50 Ω. Thus, the voltage drop across the dummy load (*V*) during pulsed transmission is used to derive the required received RL voltage during the respective continuous transmission mode with equated LVAD model (RL) power delivery.

The equation used is the following:(5)$$P=\frac{V^{2}\cdot PW}{R\;\cdot \;\left(IT+PW\right)}$$
where *V* is the amplitude voltage of the RF energy pulse measured at RL, *PW* is the width of the energy transmission pulse (ON time), *R* is the LVAD model resistance (50 Ω), *IT* is the idle time (no transmission), and (*IT + PW*) is the total cycle length or total period of the transmission waveform.

### 2.6. Coupling coil and NTC Thermistors Array Probe Arrangement

Each WPT coil surface or PLE heating probe is attached to a parallel polyethene film, of the same shape and size, with 12 adhered negative temperature coefficient (NTC) thermistors, as illustrated in Figure 3. Note that caution was taken to minimise the possible sensor wire direct RF-induced eddy current loss heating effects with the WPT system to negligible levels. The sensors had hair-thin wires (80 µm) connected to very high input impedance instrumentation amplifiers at the operating RF (200 kHz). In addition, the RF transmission efficiency difference was tested with and without the polyethene template film carrying the adhered NTC sensors array. Moreover, the NTC thermistors were calibrated before the measurement was carried out. The thermistors and the coils were placed in wet (in vitro) measurements with a known temperature, for example, water at 30 °C. Then the thermistor parameters were adjusted until the desired temperature value was obtained. These sensors have an approximately linear relationship with the temperature. However, the NTC thermistors have a resistance tolerance of +/−0.5% (equivalent to +/−0.1 °C). The Ulster University proprietary designed spiral-elliptical coils (US patent 11191973 B2) are double layer, thin (thickness ≈ 200 µm, 30 turns) and flexible and are fabricated on a polyimide substrate. The secondary coils (for channel 1 and channel 2) were implanted approximately 6–10 mm underneath the subcutaneous tissue. The primary coils were fixed above the skin and aligned with the secondary implanted coils, as illustrated in Figure 4. The following equation converts the recorded thermistor signals into temperature (*T*) in Kelvin (K).
(6)$$\frac{1}{T}=\frac{1}{T_{0}}+\frac{1}{\beta }ln\frac{R_{m}}{R_{0}}$$
where: (7)$$R_{m}=\frac{R_{0}}{\left(\frac{v_{ref}}{v_{m}}-1\right)}$$

Table 3 presents the parameters used in Equations (6) and (7), for calculating the temperature from the recoded thermistor signals.

**Figure 3 sensors-22-07775-f003:**
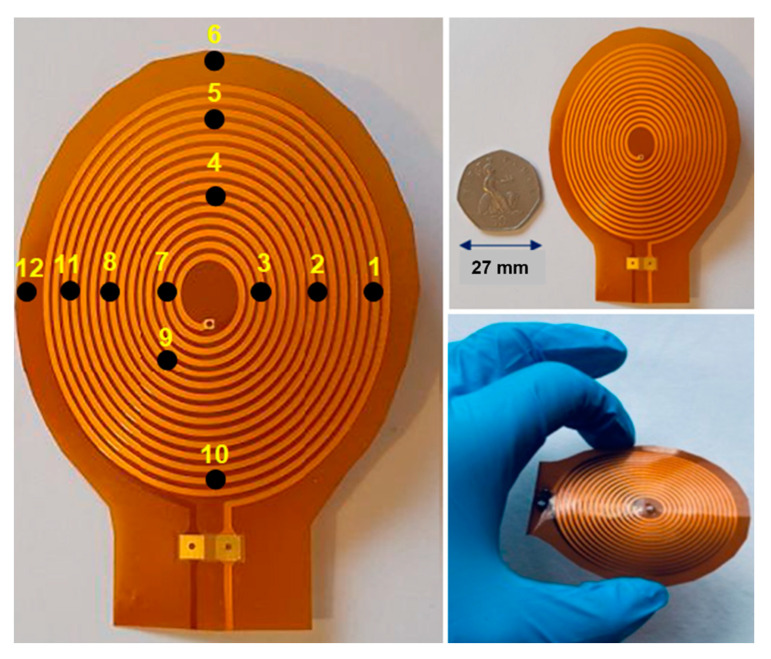
An illustration of thin and flexible coils with the position of the twelve NTC thermistors (numbered 1 to 12) adhered on the coil surface.

**Figure 4 sensors-22-07775-f004:**
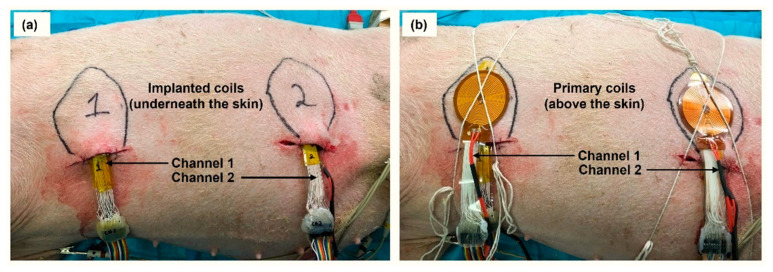
An in vivo measurement setup: (**a**) implanted coils underneath the skin, and (**b**) primary coils adhered above the skin.

**Table 3 sensors-22-07775-t003:** The parameters used in Equations (6) and (7) to calculate the temperature from the recoded thermistor signals.

Parameters	Values
$R_{0}$	10 kΩ
$T_{0}$	25 °C
$v_{ref}$	2.048 V
$v_{m}$	DAQ Channel reading as a voltage
β	temperature coefficient of the thermistor

### 2.7. In Vivo and Cadaver Studies

The in vivo measurements were carried out in 18 pig cases randomly selected (average weight 50 kg; average body temperature 37 °C, male and female) under the same measurement conditions with various power levels, pulse width and idle time (no transmission). Note that the results in Section 3 are based on the received power level for the LVADs application. The porcine study was endorsed by the Agri-Food and Bioscience Institute (AFBI) and the Animal Welfare & Ethical Review Board, and a project licence (PPL 2900) was obtained, under the Animals (Scientific Procedures) Act, from the Department of Health, the UK Home Office. Each pig was sedated (morphine, midazolam, medetomidine and ketamine by intramuscular injection), transferred to the surgical facility and a 20 g cannula placed in an auricular vein. General anaesthesia was induced with propofol, given intravenously to effect and a 9.0 mm cuffed endotracheal tube was placed.

A non-probability convenience sampling approach was chosen in order to provide the necessary data to meet the study objectives, which were to obtain a comparative thermal metrics trends assessment in limited sets of pigs per system type study with PLE and WPT. Using this sampling type approach, accurate intervals and a margin of error analysis are not formally justified.

The pulsed transmission mode variables include pulse duration, idle time and pulse amplitude voltage level, which all influence the power delivery level. In the continuous transmission mode, there is only the voltage level variable, but in this in vivo comparative study (pulsed vs. continuous), this variable is dependently adjusted to deliver the same power level achieved with the corresponding pulsed transmission mode, which was systematically run before the continuous mode run (no randomization). The continuous mode voltage level is calculated from the completed pulsed mode protocol variables using Equation (5) for matched (same) load power delivery with both modes.

For each pig case (4-h experiment), measurements were made at the lower power delivery level followed by the middle power range, and finally at high power. We performed the pulsed measurements first and then the continuous transmission measurements under the same measurement conditions. Before each measurement, there was a 2 and 5 min (no transmission) idle time to reduce the temperature to close to the baseline temperature (local heat generated due to power transmissions).

Furthermore, all the pulsed mode treatment protocols in a particular pig case were completed first; then all the matched (same power delivery level) continuous mode treatment protocols were completed in the same pig case.

General anaesthesia was maintained with isoflurane in oxygen and medical air (FiO_2_ 0.5), and the pig was ventilated (approximately 18 bpm with a tidal volume of 475 mL) to maintain normocapnia throughout. Arterial blood pressure was monitored through a 20 G cannula placed in a branch of the medical saphenous artery. Isotonic fluids were administered at the rate of 5 mlkg-1hr-1 intravenously. After skin preparation, two subcutaneous pouches were surgically created on the left side of each pig (see Figure 4a): one caudal to the elbow over the thorax and the other cranial to the stifle over the caudal abdomen. At the end of the anaesthetic period, the pigs were euthanised, without recovering from anaesthesia, by intravenous administration of a barbiturate overdose. The cadaver measurements were carried out after the pig’s clinical death.

### 2.8. Thermal Effects Metrics

The blood thermal influence (subcutaneous tissue heating mitigation effect) was evaluated by estimating the heating coefficients from the subcutaneous-coil thermistor’s temperature recording data profile with time during the WPT system power delivery to a 50 Ω pure resistive LVAD load model, or during the PLE system power loss heating emulation, both for the in vivo (alive with blood circulation) and for the cadaver (no blood circulation, euthanised model) stages of same case porcine model (pig). As described above, for the WPT system, a 12-NTC thermistor sensor array was mounted on the surface of each coil, as shown in Figure 3. For each transmission protocol, the average values of the 12 NTC sensors (in the subcutaneously implanted coil or heating probe) for each channel were used for estimating the thermal profile heating coefficient throughout the RF power transmission protocol duration (mean values with SD and SE). This amounted to a 10 to 20 min recording time. The mean temperature variations (ΔT) throughout the protocol(s) provided the tissue heating coefficient (°C/s). Thus, this study evaluates the thermal profile heating coefficients for the pulsed and continuous transmission modes to obtain evidence-based knowledge of the skin-tissue temperature cooling effect due to blood perfusion for wireless power solutions for next-generation implanted heart pumps.

### 2.9. In Silico Support Studies

Finite element analysis (COMSOL Multiphysics, ver. 5.6, Burlington, MA, USA) was used to simulate the heating effects in the subcutaneous tissue regions due to resistive loss of the coils. The simulation was performed using magnetic fields (MF), heat transfer (ht) in solids and biological media events modules and coupling of multiphysics modules. A two-dimensional axisymmetry geometry (symmetry is at r = 0) was constructed in COMSOL, as illustrated in Figure 5. Moreover, 2D axisymmetry geometry significantly reduced the computational time. The model has 149,926 (plus 34,907 internal) degrees of freedom (DOFs) to be solved. In addition, the coils and surrounding regions of coils domains and subcutaneous tissue had a higher mesh density to increase the accuracy of the simulation, in particular the region adjacent to the coils. From COMSOL multiphysics, the induction heating module and the heat transfer in biological media module were coupled to compute the temperature profile inside the subcutaneous tissue due to magnetic induction heating (RF). The frequency-transient study was set from 0–600 s at f = 200 kHz, and the MUMPS (Multifrontal Massively Parallel sparse direct Solver) direct solver was chosen, which is widely used to solve large sparse systems of linear algebraic equations. The direct MUMPS solver can utilise all of the processor cores on a single machine and store the solution out-of-core, which means the solver can offload some of the problems onto the hard disk if required and converge the solution [30]. Figure 6 shows the constructed geometry. The following bioheat equations were solved in the subcutaneous tissue to estimate the temperature [31]:(8)$$C\rho \frac{\partial T}{\partial t}=\nabla \cdot \left(K\nabla T\right)+Q_{s}+Q_{SAR}+Q_{m}-P_{b}\left(T-T_{b}\right)\;$$
where *T* is the temperature, *C* is the specific heat, ⍴ is the density of the tissue, and *K* is the thermal conductivity. The heating sources are *Q_s_*, *Q_SAR_* and *Q_m_*. The heating term *Q_s_* represents the resistive heat from the coils. We ignored *Q_SAR_* as the specific absorption rate (SAR) is significantly lower at 200 kHz. *Q_m_* is the metabolic heat source that depends on the body’s physiology, and *P_b_* is the blood perfusion. Table 4 presents the parameters used for the simulation setup. Note that parametric sweeps were used with voltage levels, pulsed width, and idle time to validate the simulation results with the in vivo measurements.

## 3. Results

This section presents the thermal profiles data obtained from the two implanted channels (channel 1 and channel 2, see Figure 4a,b). The coils or heating probe elements were placed underneath the subcutaneous tissue, and the measurements were performed in both in vivo (alive) and cadaver experimental stages of the pig model cases under same protocols. Note that results are shown based on the LVAD power rating, for example, 2.5 W, 5 W, 6 W, and 8 W received power at the dummy load, as mentioned above.

### 3.1. In Vivo Baseline and Average Temperature Measurements

An initial baseline temperature was recorded for all porcine models without transmitting any power. The measured baseline temperatures are presented in Figure 7a. The adhered 12-NTC thermistors at the implanted coil surface recorded the blood temperature as voltage signals while no power transmission was applied. The recorded voltage signals were converted into temperature (K) using Equation (6). The initial local blood temperature varied between 37.0 °C and 38.5 °C. Note that temperature varied with the position of the thermistors. Figure 7b presents a typical temperature response of the 12-NTC thermistors while the pulsed transmission protocol was applied and when recording the temperature from the subcutaneous tissue. The local temperature in the subcutaneous tissue increased from the initial baseline temperature of approximately 37.5 °C to 40 °C (ΔT = 2.5 °C). However, the temperature of the 12-NTC thermistor varied with the position on the coil surface. A transient temperature rise was observed immediately after a very short RF transmission pulse was applied (Figure 7b), followed by a transient temperature drop when there was no transmission (idle time). Thus, no significant tissue heating occurred during a complete transmission cycle (RF pulse duration **+** idle time). Figure 7c presents a typical temperature response of the 12-NTC thermistors under the continuous transmission protocol. The local temperature rises from 38.2 °C to 41 °C (ΔT = 2.8 °C). Hence, the temperature of the thermistor varies with position.

Figure 7d presents the average temperature of the 12-NTC thermistors under the pulsed and continuous transmission protocols obtained from Figure 7b,c. It is striking that the average temperature increased approximately from 38.5 °C to 39 °C (ΔT = 0.5 °C) during the pulsed transmission mode. Likewise, the temperature rose from 38.8 °C to 39.6 °C (ΔT = 0.8 °C) in the continuous transmission mode. The temperature rose while the pulse was active (pulsed transmission); however, the temperature dropped in the subsequent idle time due to blood perfusion.

On the other hand, in continuous transmission mode, the temperature rose to a certain level (depending on the applied power) and the temperature was maintained until the transmission was off. Cumulative heating could be significant in the case of continuous transmission.

It is hypothesised that blood circulation (alive model) thermal perfusion in the skin-tissue reduces the skin temperature resulting from RF-power transmission in both the pulsed and continuous transmission modes.

### 3.2. In Vivo Temperature Measurements

The in vivo temperature measurements resulting from emulated RF power loss in the implanted coils from channels 1 and 2 were recorded under both pulsed transmission protocols and their respective continuous transmission protocols, as described in the Methods section. The results show the porcine cases under the same measurement conditions. The received electric power levels were 2.5 W, 5 W, 6 W and 8 W, corresponding to the LVADs power rating levels. As mentioned above, the recorded voltage drop across the heating elements in the pulsed mode was used to derive the input voltage for the associated continuous mode by equating the delivered energy per pulsed transmission cycle (Section 2.4). Figure 8a–d present the in vivo average temperature measurements on a semi-log scale from the implanted coils for channel 1 and channel 2 for different LVADs power levels of 2.5 W, 5 W, 6 W, and 8 W.

In Figure 8a, the average temperature resulting from pulsed transmission increased from 37 °C to 37.5 °C (ΔT = 0.5 °C) in channel 1; however, in channel 2 the temperature rose from 37.5 °C to 38.5 °C (ΔT = 1 °C). It is striking that the channel 2 temperature was higher than that of channel 1. Likewise, this pattern is observed in all the recorded temperatures under both pulsed and continuous transmission mode protocols. This could be due to higher blood perfusion (cooling) in skin areas near the heart, as for channel 1. It is also noticeable that the recorded temperatures resulting from the continuous transmission were higher than those from the pulsed transmission mode, as shown in Figure 8a–d.

The maximum temperature was observed for the 8 W LVAD power level, as shown in Figure 8d. During the continuous transmission, the average temperature rose to 42 °C and 42.5 °C for channel 1 and channel 2, respectively. However, during the pulsed transmission mode, the average temperature rose to 40.5 °C and 41 °C for channel 1 and channel 2, respectively. Moreover, the average temperature was almost 1.5 °C higher in the continuous transmission mode than in the pulsed mode. From this observation, it is clear that idle time (no transmission) and blood perfusion played a vital role in mitigating the thermal effects of RF power transmission inside the subcutaneous tissue. In all the measurements, as shown in Figure 8a–d, pulsed transmission showed a lower temperature profile than continuous transmission, which would mitigate the thermal damage to the skin tissue.

The thermal heating coefficients were estimated from the average temperature data for channel 1 and channel 2 under both pulsed and continuous transmission protocols. Table 5 presents the estimated thermal heating coefficients from the in vivo measurements under the same experimental conditions. There, the estimated thermal coefficients (°C/s) show that the pulsed transmissions and continuous transmission values were similar for low power rated LVADs, for example, 2.5 W, 5 W, and 6 W power levels. However, for the 8 W load, the estimated thermal coefficient (°C/s) of the continuous transmission was at least double that of the pulsed transmission (channel 2). Moreover, the thermal heating coefficient for channel 1 was slightly higher in continuous transmission modes. This accounts for the reduced thermal effect with pulsed transmission protocols for high LVAD power rates, indicating that pulsed transmission provided more time for temperature to reduce through blood perfusion. Table 5 shows the thermal heating coefficients (mean ± SE) at various power levels of LVADs for in vivo measurements.

### 3.3. Cadaver Temperature Measurements

The thermal profile metrics data at each porcine model’s respective cadaver stage were collected for both pulsed and continuous transmission modes to investigate the tissue heating effects in the absence of blood perfusion. Figure 9 presents the measured average temperature change (ΔT) at the implanted element (channel 1) at the in vivo and cadaver stages for the 2.5 W LVAD power level. For the in vivo pulsed transmission, the average temperature rose from 39 °C to 39.5 °C (ΔT = 0.5 °C), but in the cadaver, the temperature rose from 40.7 °C to 42.1 °C (ΔT = 1.4 °C). Likewise, a similar temperature profile was observed in continuous transmission mode.

The in vivo average temperate rose from 39.4° to 40.2 °C (ΔT = 0.8 °C); however, in the cadaver measurements, the average temperature rose from 40.8° to 42 °C (ΔT = 1.2 °C). Regarding the cadaver measurements, it was confirmed that the average temperature increased in both the pulsed and continuous transmission modes. However, it is unclear why the temperature was slightly higher in the pulsed mode. The observation suggests that blood perfusion is vital in reducing the skin heating effect. 

Figure 10 summarises the average relative temperature increases in pulsed and continuous transmission modes based on 2.5 W, 5 W, 6 W and 8 W LVAD in vivo measurements. The maximum average temperature increases for various delivery power loss levels. It is noticeable that the channel 2 temperatures for both pulsed and continuous transmission presented slightly higher values than for channel 1. This could be due to higher blood perfusion (cooling) in skin areas near the heart. The temperature rose approximately from 0.5 °C to 2 °C until a 5 W load was delivered in pulse transmission. Nevertheless, continuous transmission temperature rose from 0.5 °C to 3 °C until 5 W LVADs. Moreover, the maximum temperature rose to 3.5 °C at the 8 W delivery power loss level in pulsed transmission; however, the temperature rose to 5 °C for the continuous mode transmission under the same measurement conditions. Thus, from an overall perspective, at 8 W, pulsed transmission generated 2 °C less heat than continuous transmission.

Further analysis can be found in Figure 11, which presents the absolute temperature increase in the absence of blood perfusion in pulsed and continuous transmissions mode, and absolute temperature increase in the presence of blood circulation thermal perfusion.

**Figure 10 sensors-22-07775-f010:**
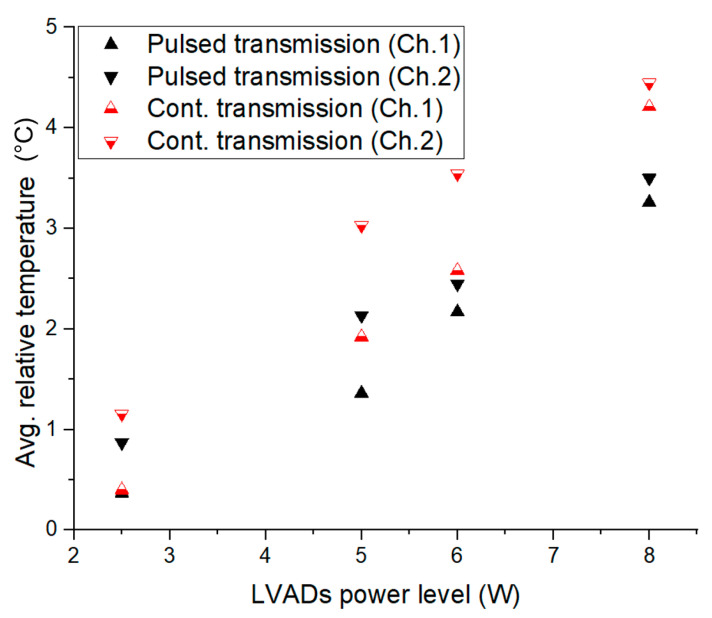
Average relative temperature change (ΔT) in the subcutaneous tissues vs. power delivered to the load in pulsed and continuous transmission protocols from the in vivo measurements with power levels of 2.5 W, 5 W, 6 W, and 8 W.

**Figure 11 sensors-22-07775-f011:**
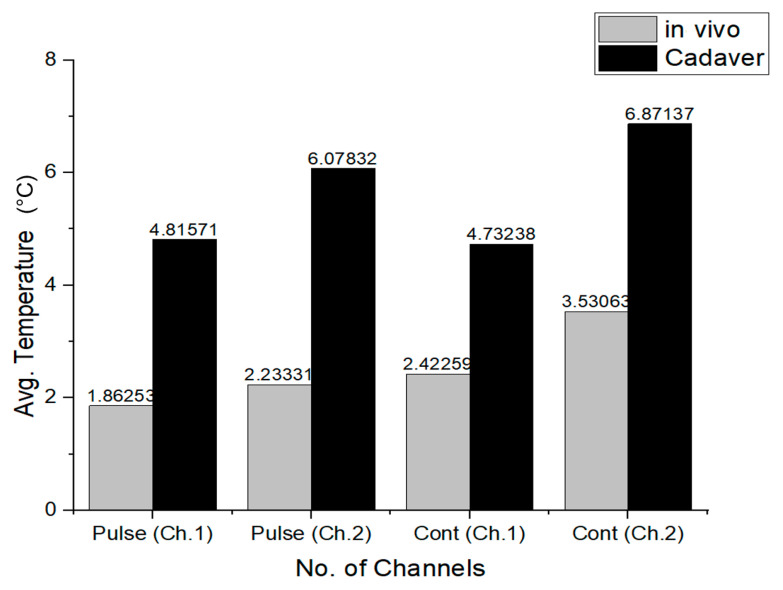
Maximum average temperature changes for in vivo and cadaver measurements at 5 W LVAD under pulsed and continuous transmission protocols (channel 1 and channel 2).

Figure 11 presents an overview of the results from this study with regard to estimated maximum average temperature change (ΔT) at the implanted element (channel 1 and channel 2) for the in vivo and cadaver stages, under the same experimental conditions for the 5 W delivery level. The in vivo mean ΔT in pulsed transmission mode (channel 1 and channel 2) was 2 °C. The mean ΔT was between 5 °C and 6 °C for the cadaver stage. The mean ΔT for continuous transmission was the highest in channel 2 (about 7 °C at the cadaver stage). In both cases, the temperatures are higher than for the pulsed transmission protocol.

### 3.4. In Silico Temperature Measurements

Finite element analysis was conducted to develop a thermal analysis model inside the subcutaneous tissue in both pulsed and continuous transmission modes in RF-coupled WPT systems. The aim is to validate the in silico model with the in vivo measurements. Note that the simulation is performed in a 2D-axisymmetry geometry, and a secondary coil (Rx) was placed underneath skin tissue, in a similar way to the in vivo measurements. As the RF pulsed and continuous energy is transmitted, current flows in the RF power coupling elements (Tx and Rx), and power loss heat is dissipated into the surrounding media. However, blood perfusion in the subcutaneous tissue reduces the temperature and protects the tissue from thermal damage. Therefore, a point is created at the top of the implanted coil surface (Figure 12b) to obtain insights into the mitigation of tissue heating by blood perfusion by visualising the point graph. The generated point graph of tissue heating inside the skin tissue is presented in Figure 12a.

From Figure 12a it is clear that the idle time (no transmission) of the pulse mode along with the blood perfusion plays a vital role in mitigating the tissue heating effects compared with the continuous transmission mode. During the pulsed transmission, the temperature rises while the pulse is active; however, the temperature drops in the subsequent idle time due to blood perfusion. In contrast, during continuous transmission, the temperature rises and maintains a higher temperature than the pulsed mode until the transmission is off. As previously mentioned, perhaps blood perfusion and idle time (no transmission) significantly reduce the temperature generated due to RF power transmission. Further comparisons of the pulse and continuous modes are illustrated in Figure 13, Figure 14 and Figure 15.

In Figure 12, Figure 13, Figure 14 and Figure 15, both (a) and (b) show the contour plots of the thermal analysis obtained from the in silico analysis in pulsed transmission modes for 2.5 W, 5 W, and 8 W LVADs power levels. In Figure 12a,b, it can be seen that the temperature varies from 37.1 °C to 38.38 °C in pulsed mode and from 37.1 °C to 39.01 °C in continuous transmission mode for 2.5 W LVAD. For 2.5 W LVAD, both transmission modes have a very low thermal profile. Perhaps, blood perfusion reduces the heat generated due to power transmission. Likewise, Figure 13a,b and Figure 14a,b show a variation in temperature from 37.1 °C to 39.19 °C in pulsed mode and from 37.1 °C to 41.17 °C in continuous mode and from 37.1 °C to 42.19 °C in pulsed and from 37.1 °C to 43.45 °C in continuous transmission modes for 5 W and 8 W LVAD, respectively. Table 6 summarises the temperature changes (ΔT) obtained from the in silico model in both pulsed and continuous RF power transmission models.

**Figure 14 sensors-22-07775-f014:**
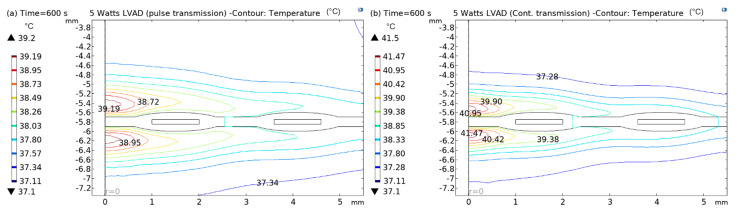
Contour plots of thermal analysis: (**a**) pulsed and (**b**) continuous transmission modes for 5 W LVAD.

**Figure 15 sensors-22-07775-f015:**
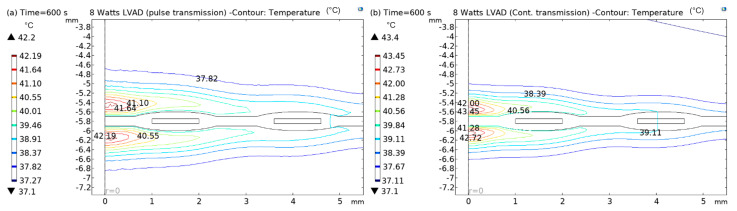
Contour plots of thermal analysis: (**a**) pulsed and (**b**) continuous transmission modes for 8 W LVAD.

It is striking that the core of the coil generates more heat than the rest of the coil area. The tissue heating is significant at core of the implanted coil. Therefore, it is expected that the current density is higher at the coil’s centre. The higher current across the centre of the coil generates more heat and dissipates it into the surrounding area, for example, skin tissue. The thermal analysis of the simulation results confirms that the temperature rises in both pulsed and continuous transmission modes; however, a higher temperature profile was observed in the continuous transmission modes at all the power levels.

**Table 6 sensors-22-07775-t006:** Summary of the simulated temperatures (ΔT) inside the skin tissue.

LVAD Power Level (W)	Pulsed (ΔT) (°C)	Continuous (ΔT) (°C)
2.5	1.28	1.91
5	2.09	4.07
8	5.09	6.35

## 4. Discussion

This work’s main objective was to investigate the impact of pulsed energy transmission mode on tissue heating effect mitigation for wireless power delivery to various power-rated implanted LVADs. The research was conducted in vivo using porcine models in two states: a living model (blood circulation) and cadaver model (control). Subcutaneous tissue thermal profile data was acquired from a thermistor sensor array around the energy transfer skin area during each experimental run for a particular power level delivery. Temperature data was then processed for quantitative analysis and the results were summarised in tables and graphs. A comparative assessment was made between pulsed and continuous transmission protocols, with various power delivery levels. Then, an in silico model was developed in conjunction with the in vivo experimental data. The analysis focus was on the idle time of the pulse transmission which was designed to minimise the tissue heating effects, aiming to prevent irreversible thermal damage to tissue. The in silico model was used to develop a thermal analysis profile was on the implanted coil and subcutaneous tissue domains to validate the in vivo measurements.

The results from the in vivo and in silico analyses summarised in Table 5 and Table 6 provide evidence-based knowledge characterising the porcine model skin tissue thermal profile in a WPT system for driving implanted artificial hearts such as LVADs. The thermal cooling role of dermal tissue blood circulation is revealed by the in vivo (alive model) study in comparison with the same case control cadaver model. Furthermore, the results from porcine studies consistently demonstrate the reduced heating effect when a using pulsed RF-transmission mode of operation versus the conventional mode of continuous RF-transmission. Figure 10 summarises the average relative temperature increases in pulsed and continuous transmission modes to drive a 2.5 W, 5 W, 6 W and 8 W LVAD for in vivo measurements. The maximum average temperature increased with each delivery power loss level. It is noticeable that channel 2 temperatures, in both pulsed and continuous transmission, presented slightly higher values than for channel 1. This could be due to higher blood perfusion (cooling) in skin areas near the heart. The temperature rose approximately from 0.5 °C to 2 °C until 5 W was delivered to the load in pulse transmission. In contrast, the continuous transmission temperature rose from 0.5 °C to 3 °C until 5 W LVADs. Moreover, the maximum temperature rose to 3.5 °C at the 8 W delivery power loss level in pulsed transmission; however, the temperature rose to 5 °C for the continuous mode transmission under the same measurement conditions. Thus, in from an overall perspective, the pulsed transmission generated 2 °C less heat than the continuous transmission for 8 W LVADs, as shown in Figure 10.

The overall results for various experimental settings are presented in Figure 11. Regarding the estimated maximum average temperature change (ΔT) at the implanted element, it was consistently revealed throughout the study that pulsed transmission [28] yields about 30% lower temperature effects than conventional continuous transmission [21]. Hence, blood perfusion factors, along with the idle time of the pulse transmission, can be harnessed to help reduce the skin heating effect using suitable WPT technology, as suggested by pulsed RF transmission waveforms proposed by the TWESMI system concept [28].

It is well known that skin heating effects are also associated with electrical inefficiency of inductively coupled coils and to the RF transmitter power switching performance [12]. These aspects were addressed in this study to further improve the efficiency of the WPT systems. Further development work would need to incorporate contingent stabilisation digital control via a transceiver link using the computer models and parametric handlers knowledge gained from the in vivo and in silico models in this study. An advanced hybrid TWESMI concept system which exploits the use of capacitive coupling methods [17] to further mitigate skin heating effects is currently under development.

## 5. Conclusions

We investigated the impact of pulsed and continuous energy transmission heating effects in in vivo and in silico models by means of novel power loss emulation methods. We used inductively coupled coil RF wireless power transmission methods supported by numerical computational methods (in silico). The study provided evidence-based knowledge about the subcutaneous blood flow cooling factors and proposed methods for harnessing this important capacity using a novel power-loss emulation system for designing safe high-power rated WPT systems with mitigated dermal tissue heating effects.

The thermal cooling role of dermal tissue blood irrigation/circulation was revealed by the study conducted in vivo (alive model) in comparison with the same case control cadaver model. Furthermore, the porcine model case studies consistently demonstrate the reduced heating effect of using a pulsed RF-transmission mode of operation versus the conventional mode of continuous RF-transmission for driving implanted artificial hearts such as LVADs. Therefore, further development of the TWESMI system concept, which inherently operates in the pulsed transmission waveform mode [28], aim to achieve longer idle times for optimising blood perfusion cooling factors.

## 6. Patents

The Ulster University proprietary design spiral-elliptical coils and RF pulsed waveform Transcutaneous Energy Transfer Systems (US patent no. 11191973 B2, granted on 07/Dec/2021; Canada patent no. 2993759; China patent no. 201680051039.3, granted on 15/Apr/2022; European patent no. EP3328490, granted on 31 August 2022).

## Figures and Tables

**Figure 5 sensors-22-07775-f005:**
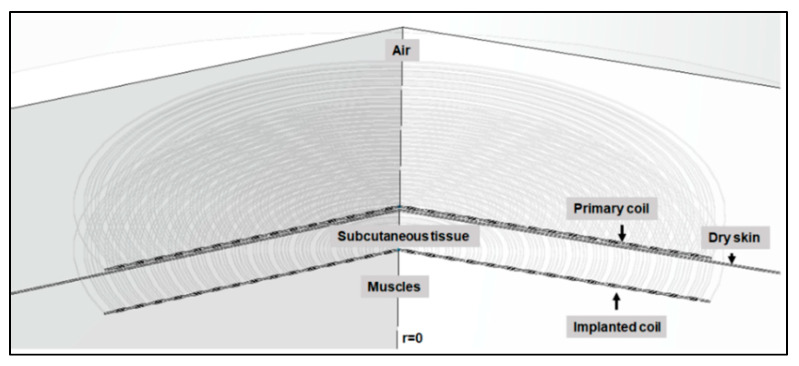
Constructed geometry in COMSOL for in silico model (Symmetry at r = 0).

**Figure 6 sensors-22-07775-f006:**
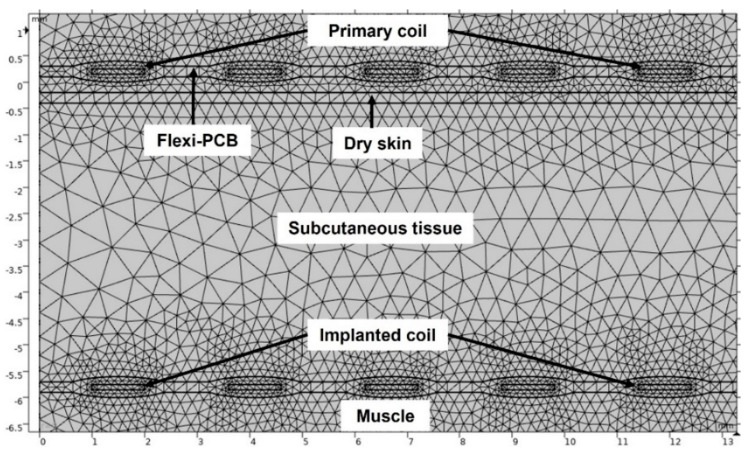
The geometry after applying the meshes.

**Figure 7 sensors-22-07775-f007:**
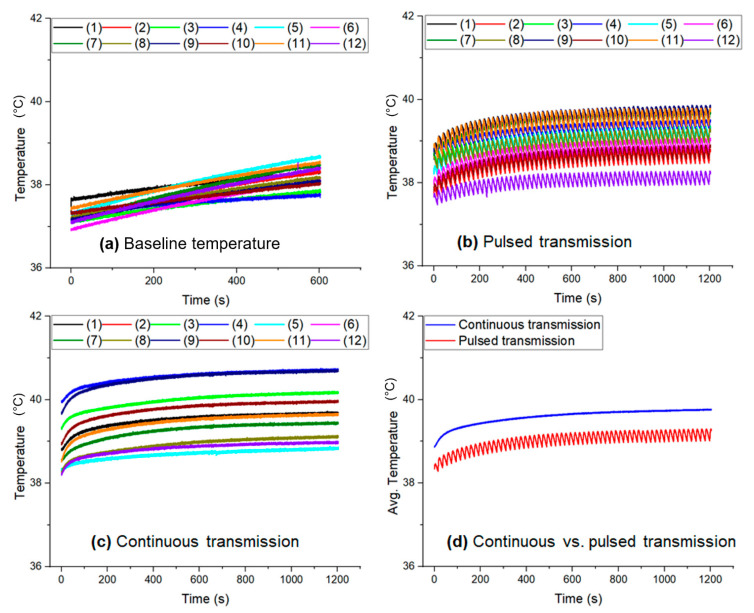
An in vivo temperature measurement from the implanted coil: (**a**) baseline temperature (power is OFF), (**b**) pulsed transmission (5 W LVAD), (**c**) continuous transmission (5 W LVAD), and (**d**) average temperature calculated from (**b**,**c**) in pulsed and continuous transmissions (for 5 W rated LVAD).

**Figure 8 sensors-22-07775-f008:**
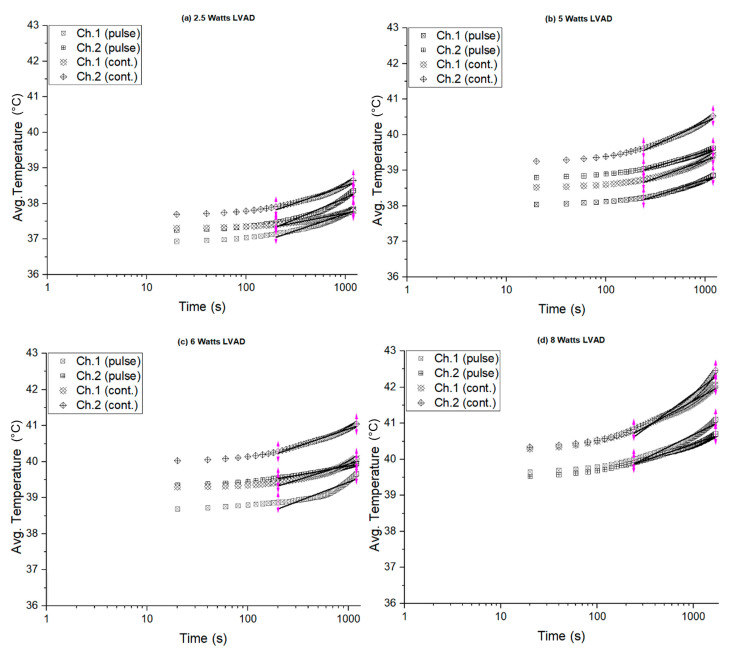
In vivo temperature measurements from the implanted coils: (**a**) 2.5 W, (**b**) 5 W, (**c**) 6 W, and (**d**) 8 W power rated LVAD (semi-log scale).

**Figure 9 sensors-22-07775-f009:**
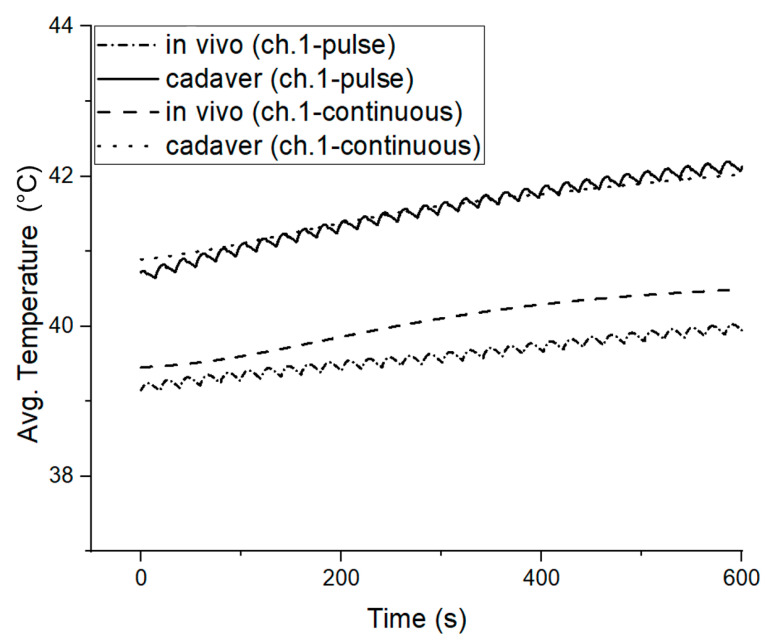
In vivo and cadaver temperature measurements from channel 1 (implanted) during pulsed and continuous transmission.

**Figure 12 sensors-22-07775-f012:**
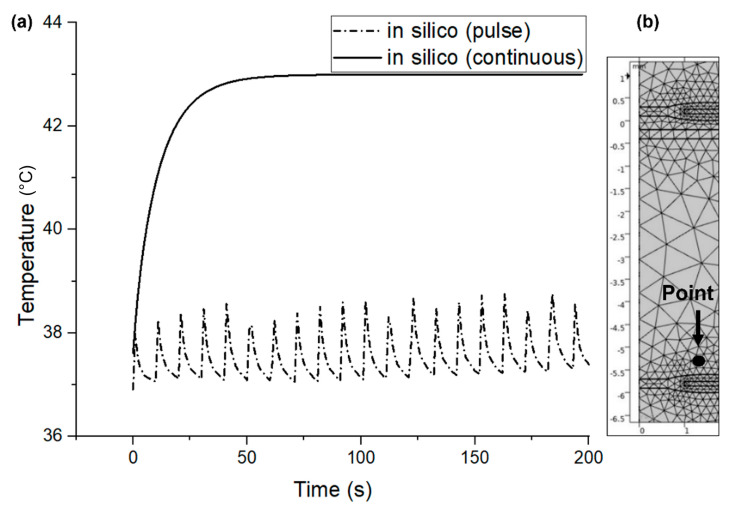
(**a**) Generated point graph from the simulated results above the implanted coil surface for 5 W LVAD, and (**b**) the position of the point.

**Figure 13 sensors-22-07775-f013:**
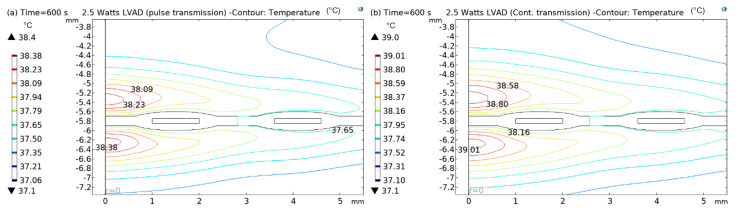
Contour plots of thermal analysis: (**a**) pulsed and (**b**) continuous transmission modes for 2.5 W LVAD.

**Table 2 sensors-22-07775-t002:** Calculated instantaneous power values using Equation (4).

Voltage, *V*_0_ (V)	Instantaneous Power, *P*_0_ (W)
50	55
75	125
100	225
125	350
150	500
175	685

**Table 4 sensors-22-07775-t004:** Parameters used for the in silico model setup.

Name	Value	Description
R_1_	5 × 10^−4^ m	Radius of the Cu track
W_1_	1 × 10^−3^ m	Width of the Cu track
H_1_	1 × 10^−4^ m	Height of the Cu track
T_d_	8 × 10^−4^ m	Inner gap between the track
⍴_b_	1 × 10^3^ kg/m^3^	Density of blood
C_p_	3639 J/(kg·K)	Specific heat of blood
⍵_p_	3.6 × 10^−3^ s^−1^	Blood perfusion rate
T_b_	37 °C	Blood temperature
f	200 kHz	Transmission frequency

**Table 5 sensors-22-07775-t005:** Thermal heating coefficients (mean ± SE) at various power levels of LVADs for in vivo measurements.

Power Levels (W)	Channel 1 (Pulsed) (°C/s)	Channel 2 (Pulsed) (°C/s)	Channel 1 (Cont.) (°C/s)	Channel 2 (Cont.) (°C/s)
2.5	0.93 ± 0.02	1.17 ± 0.04	0.53 ± 0.01	0.96 ± 0.02
5	0.91 ± 0.01	0.84 ± 0.01	0.99 ± 0.02	1.28 ± 0.02
6	1.06 ± 0.05	0.50 ± 0.08	0.86 ± 0.02	0.96 ± 0.01
8	1.30 ± 0.03	0.91 ± 0.01	1.45 ± 0.02	1.95 ± 0.04

## Data Availability

The data presented in this article are available on request from the correspondence author, and will be made available for download from the Ulster University institutional repository open facility (PURE) related to the correspondence author.

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
