# Peer review of "Transcutaneous Pulsed RF Energy Transfer Mitigates Tissue Heating in High Power Demand Implanted Device Applications: In Vivo and In Silico Models Results"

_sensors, 2022, doi:10.3390/s22207775_

Round 1

Author Response

Reviewer 1:  comments for sensors-1917051

Summary of Article:

The authors describe the methods by which they evaluated the thermal effects of perfusion and wireless power transfer parameters for LVAD applications.

    Response from authors:           ü

The authors would like to thank this reviewer for their time and interest in reviewing the initial manuscript and the revised version, after addressing the reviewer’s comments.

General Comments:

  1. Overall, the paper is well written and clearly organized. The figures and writing are of high quality. The paper offers a clear contribution given its large sample size of the porcine animal model and the heavily instrumented, temperature sensor-based experimental setup.

    Response from authors:        ü

  The authors are delighted by the reviewer’s  overall positive appreciation of the value and importance of the work presented in this manuscript.

  1. My primary suggestion is to clarify the true thesis of the article.

    Response from authors:        ü

  The overall main thesis is succinctly encapsulated in the article’s title, which has been reworded in order to help the reader to grasp the main question being addressed in the conducted study or article’s thesis: “Transcutaneous pulsed RF energy transfer mitigates tissue heating in high power demand implanted device applications”.

  • Is it that blood perfusion is the dominant driver in mitigating tissue heating during WPT?

    Response from authors:      ü

  In this study, the answer is yes, to a great considerable extent. However, to harness the heating mitigating capacity of blood perfusion, it is hypothesised that a cooling time (5 s to 120 s) is required, and that energy can be provided sustainably by transmitting high energy packs (40 J to 300 J) of short duration (30 ms - 480 ms), at about 3% duty cycle, hence, 97% of cooling cycle  (see our previously reported work [28]).To investigate this, we compared pulsed and continuous transcutaneous energy transfer  WPT operating modes and state,  and also  developed an in-silico model in conjunction with the in-vivo experimental  measurements.

  • Is it that the in-vivo animal model agrees well with the computational (in-silico) model and therefore computational models are a more economical and repeatable way to assess WPT?

    Response from authors:    ü

  To some extent this question was investigated. However, to properly answer this question, more experimental data processing and computer modelling development work is required. To clarify the in-silico model development objectives in this study, it can be said that the in-silico model development is intended to support experimental data interpretation, and to provide evidence based numerical methods for effective wireless transdermal LVAD energisation advanced practical solutions for the TWESMI system optimal operation in near future pre-clinical studies in patients undergoing LVAD treatment.

  Some of the above clarification statements are briefly mentioned in the revised Abstract Section.  

Is it that your approach for assessing power loss (“RFPLE”) is superior to other methods?

    Response from authors:        ü

  We have renamed the “RFPLE” acronym to PLE (Power loss Emulating), to reduce confusion caused by the RF letters. The PLE approach does not substitute any WPT system. So, the answer is no, it is not a superior method. The PLE system tool emulates heating power loss, as a result of some method of transcutaneous energy transfer: could be ultrasound, optical, capacitive coupling, inductive coupling, etc. Thus the PLE system is for a generic study of tissue thermal profile experimental data gathering, and the impact of blood perfusion in the in-vivo thermal data.

       Thus, the PLE approach offers some versatility advantages. The PLE prototype used in this study was validated in the comparative assessment of transmission modes: Pulsed vs Continuous, in the overall conclusion that pulsed transmission can harness blood perfusion for heating mitigation; same conclusion using the RF coupled coil WPT system in separate sets of 6 pigs.

  • Is it that cadaver models are unsuitable for any assessment of heating effects from WPT? Etc.? Many other such statements seem to be supported, yet the new and most dominant contribution is somewhat unclear.

    Response from authors:       ü

  The cadaver model final stage at the end of each pig case, provided important quantitative evidence about the significant role of blood perfusion in mitigating heating effects when pulsed energy transmission mode is used; which allows a certain cooling time. The cadaver model provided two important supporting evidenced results: 1)   In the absence of blood circulation, the skin heating effects are absolutely higher, 2) there is no clear difference in heating effects regardless of the energy transmission mode (Pulsed or Continuous).

       To help clarify the main questions addressed in this study, If for instance, the practical energy transfer efficiency is 80%, then, to deliver 10 W to the implanted load, 2W will be constantly dedicated on heating the 5cm diameter disk of subcutaneous tissue (a lot!). Then, how much can blood perfusion action help to cool the heating effect (2W)? Is it the same if the energy is transferred in periodic high energy packs, allowing an idle cooling time? We addressed this question using two ways: 1) using a Power Loss Emulating (PLE) system, providing DC ohmic heating (e.g. 2W continuous), either continuously, or pulsed with same net energy per cycle period (also 2W); 2) using an inductively coupled coils WPT operating at RF (200 kHz), with coupling inefficiency power loss heating, either at continuous transmission or pulsed transmission (to deliver 8W at the implant device load).

        The las paragraph of the Conclusions Section, briefly presents the dominant contribution of the studies.

  1. The experimental methods could be clarified in terms of the “treatment” variables and their manipulation strategy. Are power transfer and pulsed vs. non-pulsed the primary variables of interest in your study?

    Response from authors:      ü

   Yes, power transfer (to implanted load: LVAD) and pulsed vs. continuous transmission modes, are the primary variables of interest in this study.

       This study investigated the pulsed transmission mode, which has 3 characterising variables to play with: amplitude voltage level, pulse width and idle time (no transmission), for managing a chosen load (LVAD) power level (see Table 5): 2.5 W, 5 W, 6 W and 8 W. Characterising variables were investigated for understanding the best approach to mitigate the tissue heating effect, which is a critical challenge and limitation with conventional continuous transmission WPT technology.

  • What role does channel play?

     Response from authors:        ü

  Clarifying sentences have been added at the 3rd paragraph of the Introduction Section and in the 1st paragraph of Section 2. Basically, the 2-channel systems approach for the in-vivo studies would offer the additional advantages of increasing (doubling) the independent experimental data gathering for skin tissue thermal profile characterisation, and also for providing a real experimental in-vivo setting for validating the TWESMI multichannel concept, with each of the two channels at different and distant skin areas in the body, and delivering electric power to a single device load (LVAD model).

  • Your variables seem to include pulse duration, idle time, voltage, which all influence power delivery. Thus, power is a dependent variable confounded with all of the preceding variables. Perhaps this is not an issue in achieving your experimental goals, but the confounding (internal correlation) prohibits conclusions related to the effects of the individual variables, which is not

    Response from authors:       ü

   Yes, in the pulsed transmission mode, the variables include pulse duration, idle time and pulse amplitude voltage level, which all influence power delivery level. In the continuous transmission mode, there is only the voltage level variable, but in this comparative study (pulsed vs continuous), this variable is dependently adjusted to deliver the same power level achieved with the corresponding pulsed transmission mode ran always before the continuous mode run; the continuous mode voltage level is calculated from the pulsed mode variables using Eq. (5), for matched (same) load power delivery with both modes.

    And yes, load power delivery is a dependent variable confounded with all of the respective transmission mode variables.

    Section 2.7 has been revised to include these experimental procedures.

  • Moreover, use of randomization was not discussed. For example, what treatment was applied first – pulsed or continuous? Was this chosen randomly for each animal or were all animals given the pulsed treatment first? If all animals received treatments in the same order, this ordering is now confounded with the pulsed versus continuous effect. In the case of identical order, one could ask if the data are affected by permanent changes in the tissue that occur as soon as any WPT of sufficiently high power is applied.

    Response from authors:    ü

    To address these particular questions, we have revised Section 2.7; first stating the random selection of the pig for each experiment case, and also added a brief paragraph about the adopted sampling type approach for this study: non-probability convenience sampling, in some of its implications.

      As mentioned in the previous point, the pulsed transmission protocol was always applied first. But not immediately before the continuous treatment; all pulsed treatment protocols in a particular pig case were completed first, then, all matched (same power delivery level) continuous treatment protocols were completed after, in the same pig case. Time consuming instrumentational connections setup is required for changing the transmission mode from pulsed to continuous. Thus, in this aspect of the study, randomisation was not applicable; a systematic order had to be applied, as further clarified in the following comment point.

  • The same comments apply for power level. If power levels were completed in random order, the results may be different than if the power level was escalated identically for each animal. Please consider discussing these basic experimental design considerations in Section 2.

    Response from authors:    ü

    The low-power delivery level transmission protocols experimental data gathering were completed first, then middle-power range, and finally the high-power levels. We implemented pulsed transmission measurements first and followed (not immediately after) by subsequent continuous transmission measurements under the same conditions. Before each experimental protocol for a set power delivery level, there is 2 and 5 min (no transmission) idle time to reduce the skin temperature close to the baseline temperature (local heat generated due to power transmissions).

  Section 2.7 has been revised to include these experimental procedures.

  1. The use of the term “RFPLE system” seems to imply the design of some in vitro, repeatable, benchtop system to assess the effects of WPT in LVAD applications. However, this coined term seems to be synonymous with “instrumented porcine animal ” I would suggest removing the RFPLE terminology as it adds very little to the paper in my opinion.

    Response from authors:   ü

  The authors are thankful for the valuable reviewer’s perceived view on the use of the “RFPLE” acronym in our work manuscript, and for their related comments and suggestions for revising some needed clarification to improve understanding of our work by the general reader of the intended article.

       For this, we have revised the first 12 lines of the Abstract Section and the first 3 lines of Section 2.1, to clarify about our new method device approach (now given the  PLE acronym, to avoid confusing the reader with the RF letters) to study the skin tissue thermal profiles under two main energy transfer modes to supply the same load power rate level: a) Pulsed high energy packs transfer, causing pulsed heating effects on the transcutaneous energy coupling area, and b) Continuous low energy transfer (the conventional way).

        Inefficiency of the energy transmission process; either by RF inductive coupling, or RF capacitive coupling [Ref. 17], or perhaps, some ultrasound energy or some optical energy transfer method, etc. All of these have energy transfer inefficiency that translates into skin heating; either a) pulsed heating, or b) continuous low intensity heating. Thus, regardless of the source of skin heating in transdermal energy transfer, the heating effect can be emulated simply by ohmic heating under either a) DC high energy electric pulses, b) DC continuous low energy ohmic heating, the level of which depends on the load (LVAD) required power drive.

    To further clarify the reviewer’s particular understanding, the PLE system is not a standalone bench based reproducible experiment, and it does not model the living skin and blood perfusion in-vivo experiment; it is solely dedicated to gather experimental thermal profile data in the in-vivo studies in a porcine living or cadaver model study in a generic way.

Specific Comments:

  1. Line 34: “second arm of the study” implies a type of randomized clinical study, which is not the case. I would recommend removing this terminology.

    Response from authors:    ü

 The terminology “second arm of the study” has been removed.

  1. Line 41: “placebo” does not seem like an appropriate synonym for I recommend not using this word in this context.

    Response from authors:    ü

 The term “placebo” has been removed.

  1. Sentence on line 79-80 “It has been well reported…” is awkwardly Consider re-wording.

    Response from authors:     ü

 The line 79-80 has been re-worded as suggested.

  1. The power levels listed on Line 136 seem to contradict the implication that you are using 12V batteries. For example 4, 12 V batteries in series is 48 V, not 50V, 72 V is also possible through series connections, but not 75 V, etc. Please clarify.

    Response from authors:     ü

 Each lead acid battery has a nominal voltage of 12 V, as a device. However, in practice, when it is at full charge, the battery can have a maximum voltage value of about 13.2 V (at the start) and a minimum of 11.8V (at the end) during the time of the experiment, hence, an estimated average value of 12.5V as representative voltage level indication. For this reason, in Table 1, we have written the level representative value of 50V (not 48 V) for when we use four batteries.

       In section 2.2, the 12V has been replaced with the more realistic representative value of 12.5V, with a clarification statement, and in Table 1, we have corrected a typo to 75V, in the 2nd row of parameters.

  1. In Section 6 or near Equation (6), consider discussing your calibration techniques or if you were relying solely upon the manufacturer’s data for the thermistors.

    Response from authors:     ü

   The negative temperature coefficient (NTC)  thermistors were calibrated before the measurement was carried out. The thermistors and the coils were placed in wet (in-vitro) measurements with a known temperature, for example, water at 30 °C. Then the thermistors parameter is adjusted until getting the desired temperature value. These sensors have an approximately linear relationship with the temperature. However, NTC thermistors have a resistance tolerance of +/-0.5% (equivalent to +/-0.1°C). In each porcine case, we also did baseline measurement for calibration purpose.

    In Section 2.6, a discussion of the calibration method has been added and also a new Table 3 presenting parameter values used in Eq. (6) and Eq. (7) to calculate the temperature from the recorded thermistor signals.           

  1. The variable V0 does not seem to be defined in Equation (4) and it is unclear if it represents the same voltage as used in Equation (5).

    Response from authors:           ü

 The variable V0 is the applied indicative PLE system input voltage level. We have now clearly defined it in Section 2.4. Also, please see variable V0  in  Table 2. The voltage V0  in Equation (4) is not the same voltage variable V in equation (5);  the latter, corresponds to the voltage measurement across the output dummy load (50Ω), as explained in Section 2.5.

  1. The Vm variable in Equation (7) seems to be lowercase, yet it is capitalized in Equation (11). Further, Equations (8) through (12) seem better suited to a table.

    Response from authors:       ü

  The variable vm is written in lowercase. Eq. (8) to Eq. (12) have been edited into a new table. Please see Table.3.

  1. In Section 9, you mention refined mesh density near the coils and subcutaneous tissue. Did you complete any type of mesh convergence study to see if further mesh refinement or coarsening had any influence on the output?

    Response from authors:     ü

  Yes, we did a higher discretization (smaller mesh) convergence study. A smaller mesh increases the computational time. A uniform tiny mesh size everywhere leads to unachievable computing power requirements, and convergence of the simulation is one of the major issues. However, the coarse mesh dramatically reduces the computational time, and the result's accuracy is not satisfactory. Considering all these issues: higher discretization (smaller mesh) is chosen near the coils and subcutaneous tissue. We are interested in estimating the temperature at the coil's surface and next to the subcutaneous area (not into the other domain).  

  1. Possible typo: Line 293: “heart transfer” should be “heat transfer”?

    Response from authors:       ü

 The typo correction has been done, thanks.  

  1. Beginning of line 339: wording is Consider re-wording.

    Response from authors:        ü

  Agree. The first 4 lines of Section 3.1, including the indicated awkward line, have been re-worded.

  1. In some of your graphs, consider removing color printing or color viewing dependence by using dashes or different marker types. For example, Figure 9 could benefit from dashed lines to distinguish between pulsed and continuous.

    Response from authors:       ü

  Ok. To address this important comment, most of graphs, including Fig. 9, have now been presented in mono-colour; using dashed lines, etc.

  1. Consider moving the in-silico section (3.4) before the experimental section. The usual paradigm is to “validate” (word used in line 456) the numerical model with experiments and not vice versa as you seem to have done. Thus, at a minimum, I would consider re-wording the opening sentence of Section 3.4.

    Response from authors:       ü

    Thank you for this suggestion to improve the manuscript. To address this comment, the authors have opted to re-word the opening sentence in Section 3.4. This action was considered to be more manageable while minimising modifications in the revised manuscript; to be less confusing to other reviewers.

  1. What are the limitations of the “thermal heating coefficient” parameter listed in Table 4? The temperature rise versus time is nonlinear and would eventually For example, at what point do other sources of heat transfer become relevant (e.g., melting phase change of subcutaneous fat?) Is this a common parameter that others have measured using your same protocol? If so, more references may be required to support this.

    Response from authors:       ü

  The main limitation is that cumulation heating occurs- as a result, the temperature goes up, although there is no transmission (idle time) in pulse transmission mode. The objective of estimating the thermal heating coefficient is to find any linear relationship between treatment time, for example, 20 min and the impact of blood perfusion cooling effect due to local heat generated by the power loss of the implanted coil (for WPT) or heating element (for PLE), in a comparative assessment to evaluate thermal effects differences under same environment, pig case and electrical load conditions. Also, we observed how effectively temperature reached a plateau as a factor of time (in-log scale). In a nutshell, for high power rated LVAD -The heating coefficient (°C/s) is higher- which can be interpreted as high energy dissipating more heat. However, blood perfusion and idle time (no transmission) of the pulsatile transmission will prevent tissue from thermal damage. Suggested related references:  [28, 29]. Some related discussion is presented in the 2nd and new 3rd paragraphs of the Section 4 (Discussion).   Due to the new added Table 3, your referred Table 4 is now Table 5.

  1. Figure 10 is not particularly helpful and can be described in a single I would recommend removing it.

    Response from authors:      ü

  Agree. The original, above referred Fig. 10, has now been deleted. Please note that in addressing another reviewer’s comment to bring forward the original figures 14 and 15 into the Results Section, these to are now Fig. 10 and Fig. 11, respectively.

  1. Please consider elaborating on the importance and significance of the two spatial channels (probe pairs) in your This seems to be an experimental variable whose effect is not discussed extensively.

    Response from authors:     ü

  The authors much appreciate this complementary observation and suggestion. To address this comment-point some clarifying sentences have been added at the 3rd paragraph of the Introduction Section and in the 1st paragraph of Section 2, as copied below. Basically, the TWESMI idea is conceived as a multichannel system (minimum of 2 WPT transcutaneous coupling point placed on the anterior torso body surface), with the main advantages to:

   “… a) reduce the overall coupling energy density (a 4-channel system would present less skin heat for a fixed load power level), b) to offer higher operational reliability by redundancy (in case a transfer channel fails), and c) to enable capacitive coupling (using plates; not coils (see Ref. [17]), and only possible with an even number of channels (minimum two channels).”

   “ … The 2-channel systems approach for the in-vivo studies would offer the additional advantages of a) increasing (doubling) the independent experimental data gathering for skin tissue thermal profile characterisation; thus, a set of 6 pig cases (during one week) could provide the same experimental data as of 12 pig cases (requiring more resources, and during two weeks) with 1-channel WPT systems and in a shorter research time, and b) providing a real experimental in-vivo setting for validating the TWESMI multichannel concept, with each of the two channels at different and distant skin areas in the body, and delivering electric power to a single device load (LVAD electrical model).

____________________________________________________________________________

Reviewer 2 Report

Interersting paper quantifying heating of tissue during wireless power transfers to power medical implants and suggesting pulsing power transfer to mitigate tissue heating. The language could be improved: very long sentences which are hard to follow, and some inconsistensy with naming. There are some results which are not clearly described, and the results and discussion section is not clearly divided (might be merged fully or separated more). 

Detailed comments:

1. Introduction is clear and well written

2. In fig 2, mark the transmitter (Tx) and the reciever (Rx) for clarity.

3. Line 173,p5: Is the WPT not a part of the RFPLE system? This is unclear, please mark WPT in figure 1, or explain if WPT is not part of the RFPLE system and show the full WPT system as well. Clarify and use consistent language to avoid confusion.

4. Equation 4: V0 and Rcoil are not defined - please define.

5. Line 204: sentence long and some words are missing, please rephrase and shorten sentence into several sentences.

6. Fig 3: 27cm is very large for a penny, do you mean 27mm?

7. Line 265: write figure 4 a) instead of figure 4 left.

8. Line 295: what is MUMPS? please describe and add reference.

9. Line 331: Coils or heating probe elements? Is this the same thing or two different things? Please clarify. I think you might mean that there are heating probe elements on the coils? Or did you actually do measurements with the coils alone and the heating probe elements alone?

10. Line 338-339: This are methods and already explained. I think you can remove them and also shorten the results section many other places by removing method already described in the method section.

11. Figure 7a: why is there a rise in temperature during baseline measurements?

12. Subsection 3.1 and 3.2 are both in-vivo results, right? Please update subtitle 3.1 to clarify this.

13. Line 416-418: Almost similar? In table 4 there is many probably significant differences (small standard deviation and different averages). For example for 2.5W ch1 pulsed is 75% higher than c1 continuous. Did you do statistics comparing continous to pulsatile WPT? 

14. Table 4 is also given in figure 8. Please remove one of them to not duplicate the results. Remove from figure?

15. Line 451: malpositioned sentence. The sentence after starting with 'Thus' points back to the sentence before, such that the Thus sentence now does not make sense because it points to line 451 which is about something else. Please modify these sentences. 

16. Line 452: The jump to simulations data is not clear. Why do you refer to figure 13? Did you mean figure 14? This figure should maybe be added to the results instead of the discussion section.

17.  Figure 12 and 13: why did you use different color legend? Please use the same coloring and maybe the two figures could be merged? Adding 13 a below 12 a and 13b below 12b etc for easier direct comparison? In the current state the figures are very hard to compare. 

18: Discussion section. Please start with a short recap of your study aims and objectives and what have been investigated before discussing results. 

19. Do not introduce new results in the discussion section. Figure 14 is new results not shown in the results or they are results from Figure 8? Maybe describe the results in the results section and discuss them in the discussion section. 

Author Response

Reviewer 2: Response to Comments and Suggestions for sensors-1917051

Interesting paper quantifying heating of tissue during wireless power transfers to power medical implants and suggesting pulsing power transfer to mitigate tissue heating. The language could be improved: very long sentences which are hard to follow, and some inconsistency with naming. There are some results which are not clearly described, and the results and discussion section is not clearly divided (might be merged fully or separated more). 

    Response from authors:     ü

   The authors are most grateful for this reviewer’s interest and work on reviewing the original manuscript and the resubmitted revised version.

     The language in the revised manuscript has been revised throughout and many edits have been done for this aspect of the reviewer’s comments.

    The Results and the Discussion Sections have been revised and new clarifying paragraphs added; some results Figures have been replaced or moved forward from the Discussion section. A new Table has been added.

Detailed comments:

  1. Introduction is clear and well written.

    Response from authors:      ü

   The authors much appreciate this positive opinion feedback about the Introduction Section, thanks.

  1. In fig 2, mark the transmitter (Tx) and the receiver (Rx) for clarity.

    Response from authors:     ü

    Agree. Therefore, Fig. 2 has been amended accordingly; clearly inserted the Tx and Rx legends.

  1. Line 173,p5: Is the WPT not a part of the RFPLE system? This is unclear, please mark WPT in figure 1, or explain if WPT is not part of the RFPLE system and show the full WPT system as well. Clarify and use consistent language to avoid confusion.

    Response from authors:     ü

   The RF inductively coupled WPT system and the Power Loss Emulation (renamed acronym to PLE, to avoid confusion with the RF operated WPT) are separate and different system devices for two different experimental in-vivo approaches for investigating the in-vivo heating effect and blood perfusion thermal profile, and these two different systems are used in separate sets of pigs, in a different experiment time (weeks in between).

    We have revised the first 12 lines of the Abstract Section and the first 3 lines of Section 2.1, to clarify about our new method device approach (PLE), for the study of in-vivo skin tissue thermal profiles under two main energy transfer modes to deliver the same load power rate level: a) Pulsed high energy packs transfer, causing pulsed heating effects on the transcutaneous energy coupling area, and b) Continuous low energy transfer (the conventional way).

   The full inductively RF coupled (operated at 200 kHz) WPT system block diagram is depicted in Figure 2.

  1. Equation 4: Vand Rcoilare not defined - please define.

    Response from authors:     ü

    Noted. We have now defined variables V0 and Rcoil used in Eq. (4), in subsection 2.4, in the revised manuscript.

  1. Line 204: sentence long and some words are missing, please rephrase and shorten sentence into several sentences.

    Response from authors:    ü

      Noted. The long sentence in subsection 2.5 initial 5 lines,  have been shortened and rephrased in the manuscript revised version.

  1. Fig 3: 27cm is very large for a penny, do you mean 27mm?

Response from authors:    ü

  Noted. Yes, it should be 27mm. We have fixed that typo in Fig. 3.

  1. Line 265: write figure 4 a) instead of figure 4 left.

    Response from authors:

 Yes, written as Fig. 4(a).

  1. Line 295: what is MUMPS? please describe and add reference.

    Response from authors:

    Noted.  To address this comment, subsection 2.9 has been revised, and some additional lines have been edited to describe and clarify this and two respective references were added.

       MUMPS (MUltifrontal Massively Parallel sparse direct Solver) is used to simulate the thermal profile. MUMPS is a wider used to solve the large sparse systems of linear algebraic equations. The direct MUMPS solver can utilise all of the processor cores on a single machine and store the solution out-of-core, which means the solver can offload some of the problems onto the hard disk. The MUMPS solver is fully coupled. The reference are available in COMSOL user manual and online blogs (Support)[30].  

  1. Line 331: Coils or heating probe elements? Is this the same thing or two different things? Please clarify. I think you might mean that there are heating probe elements on the coils? Or did you actually do measurements with the coils alone and the heating probe elements alone?

    Response from authors:     ü

   Coils and heating probes are two different things. Coil element probes are used with the RF inductive coupling WPT system, as described in subsection 2.2. Ohmic heating element probes are exclusively used with the Power Loss Emulating (PLE) system, as is now clarified in the revised subsection 2.1. These two systems are used alone (never in a combined way) in separate experiments with separate set of pigs, and usually in different days. We have revised subsection 2.1 to address this reviewer’s comment.

  1. Line 338-339: This are methods and already explained. I think you can remove them and also shorten the results section many other places by removing method already described in the method section.

    Response from authors:     ü

  The sentence with  ‘already  described method’ has been removed and reworded as follows:

“The adhered 12-NTC thermistors at the implanted coil surface recorded the blood temperature as voltage signals while no power transmission was set up.”

  1. Figure 7a: why is there a rise in temperature during baseline measurements?

    Response from authors:     ü

   During the baseline measurement, there is no power ON. The baseline measurement was taken at the beginning after the surgery was done. Perhaps, the pigs are not settled before the surgery. This might be one of the reasons for to increase in the baseline temperature. In the surgery room, a heating system was on to raise the room temperature due to the porcine study being carried out during the winter. The measurement time is 10min. The measured temperature varies with the position of the thermistor on the coil surface. If we could average all 12 thermistors, the average temperature would eventually become a plateau. However, all the blood vessels inside the subcutaneous fat deflect from the coil surface.

      Moreover, a 20min baseline is preferable to get the plateau; however, we could not do this due to the limited time for the porcine study. Also, we need to consider the noise and interface with other devices in the lab.

  1. Subsection 3.1 and 3.2 are both in-vivo results, right? Please update subtitle 3.1 to clarify this.

    Response from authors:     ü

   Agree. Yes, both sections are in-vivo experiment results. Thus, we have updated subtitle 3.1 to be more clear: “In-vivo baseline and average temperature measurements”.

  1. Line 416-418: Almost similar? In table 4 there is many probably significant differences (small standard deviation and different averages). For example for 2.5W ch1 pulsed is 75% higher than c1 continuous. Did you do statistics comparing continuous to pulsatile WPT? 

   Response from authors:     ü

  The then Table 4 (now is Table 5) presented the thermal heating coefficient (°C/s) with various power rated load levels (LVAD), from the averaged temperature of six porcine cases under same measurement conditions.  Although the heating coefficient varies along with the power levels, pulsed and continuous transmission modes, but in low power LVAD- the heating coefficient does not increase a lot (insignificant effect on heating), in both the transmission modes. However, for high power LVAD- continuous modes generated more heats than the pulse mode. No, we did not perform statistical significance analysis (just calculated simple statistics: mean and SD).

  1. Table 4 is also given in figure 8. Please remove one of them to not duplicate the results. Remove from figure?

    Response from authors:       ü

    Noted. Hence, we have removed the Table 4 content within Figure 8.

  1. Line 451: malpositioned sentence. The sentence after starting with 'Thus' points back to the sentence before, such that the Thus sentence now does not make sense because it points to line 451 which is about something else. Please modify these sentences. 

    Response from authors:       ü

    Agree. Hence Fig. 13 and 14 are moved to the results section (before the in-silico results), and they have become Fig. 10 and Fig. 11, respectively. The malpositioned sentence ther has been reworded to make sense now.

  1. Line 452: The jump to simulations data is not clear. Why do you refer to figure 13? Did you mean figure 14? This figure should maybe be added to the results instead of the discussion section.

    Response from authors:       ü

  Noted. Both Fig. 13 and 14 are moved to the results section (before the in-silico results). Now, they are Fig. 10 and Fig. 11 respectively. The related text typo there has been corrected to refer to Fig. 11 (same old Fig. 14 ).

  1. Figure 12 and 13: why did you use different color legend? Please use the same coloring and maybe the two figures could be merged? Adding 13 a below 12 a and 13b below 12b etc. for easier direct comparison? In the current state the figures are very hard to compare.

    Response from authors:   ü

   Noted. Therefore, we have removed the different colours legend and updated the figures into a simple contour plots. Thus, now is easier to visualise the temperature from the contour lines with the labels.

  1. Discussion section. Please start with a short recap of your study aims and objectives and what have been investigated before discussing results. 

    Response from authors:     ü

   Agree. A short recap paragraph has been edited at the start of the Discussion Section.

  1. Do not introduce new results in the discussion section. Figure 14 is new results not shown in the results or they are results from Figure 8? Maybe describe the results in the results section and discuss them in the discussion section. 

    Response from authors:     ü

    The two results related figures in the Discussion Section were moved  to the Results Section (now  Fig. 10  and Fig. 11). Additional discussion paragraph in the Discussion is about Fig.11.
